# Addressing Dual Disorders in a Medium-Term Admission Unit

**DOI:** 10.3390/brainsci12010024

**Published:** 2021-12-26

**Authors:** Francisco Arnau, Ana Benito, Mariano Villar, María Elena Ortega, Lucía López-Peláez, Gonzalo Haro

**Affiliations:** 1Department of Medicine and Surgery, Universidad Cardenal Herrera-CEU, CEU Universities, 12006 Castelló de la Plana, Spain; anabenitodel@hotmail.com (A.B.); ortegaelena96@gmail.com (M.E.O.); gonzalo.haro@uchceu.es (G.H.); 2Consorcio Hospital Provincial de Castellón, 12002 Castelló de la Plana, Spain; mvillar15@gmail.com; 3Mental Health Unit, Hospital General Universitario de Valencia, 46900 Torrente, Spain; 4Hospital Álvaro Cunqueiro, 36213 Vigo, Spain; lucia.lopezpelaez.hamann@gmail.com

**Keywords:** serious mental disorder, medium-stay unit, dual disorder

## Abstract

Due to the significant functional repercussions suffered by patients with dual disorder, we must evaluate which ones can benefit from intensive rehabilitative therapies in medium-stay psychiatric units. This was a retrospective study of patient medical records which intended to analyze sociodemographic and clinical variables and parameters related to the hospitalization and discharge of patients admitted to the Medium-Stay Unit (MSU) at the Castellón Provincial Hospital Consortium over 2 years (2017 and 2018), according to the presence or absence of dual disorders in these patients. Patients with a dual disorder represented 55.2% of the hospitalized patients. A higher proportion of them were male, were relatively younger, and had an earlier onset of mental illness, fewer associated medical pathologies, and shorter hospital admission times to the Short-Term Hospitalization Unit than those who did not present a dual disorder. A diagnosis on the schizophrenia spectrum with cannabis use or polyconsumption was the most common diagnosis; 98.2% of all the patients responded adequately to admission to the MSU. This work highlighted the need for higher doses of depot paliperidone in patients with dual disorders.

## 1. Introduction

According to the international meta-analysis by Steel et al. [1], mental disorders are highly prevalent worldwide. This agrees with the latest data published by the Spanish National Health System in 2020, in which mental health was classified as the sixth most chronic health problem dealt with by the Spanish healthcare system [2,3]. Dual Disorders (DDs), defined as the concurrence of both a mental disorder and a substance use disorder, are prevalent and represent a therapeutic challenge given that these patients tend to have a greater functional impact and reduced therapeutic adherence [4]. In that sense, the use of long-lasting injectable antipsychotics as paliperidone was associated with a significantly lower rate of treatment failure, and with fewer all-cause and substance use related inpatient admissions or long-term care stays and greater medical cost savings [5].

Mental health rehabilitation services were established during the era of deinstitutionalization in the latter half of the last century. Since then, their focus on ‘resettlement’ of the residents of asylums into community-based settings has evolved, as it became increasingly clear that most individuals had the capacity to gain (or regain) skills that allowed them to live and participate with increased independence in the community. With the gradual expansion and greater specialization of community-based mental health services over recent decades, contemporary mental health rehabilitation services have increasingly focused on people with more severe and complex problems. However, their remit is not always clear and varies in different settings [6]. In Spain, as in different countries around the world, there are a heterogeneous development by regions of resources available to psychosocial rehabilitation people with severe mental disorders. For some people with severe and persisting mental illness, some regions have developed First Housing programs, Community-Based Residential Care, and the more recent Transitional Residential Rehabilitation approach. Within the latter model, Medium-Stay Units (MSUs) can be included [7].

MSUs are intended to provide comprehensive intensive rehabilitative treatment, with the aim of minimizing chronicity and promoting greater autonomy in people with severe and persisting mental illness [8]. Given the limited number of publications and the importance of examining the characteristics of DD patients in any psychosocial rehabilitation service [9], we aimed to analyze the possible differences according to the presence or absence of DDs in patients admitted to the MSU at the Castellón Provincial Hospital Consortium (CHP in its Spanish initialism), including an analysis with a gender perspective.

## 2. Materials and Methods

This was a retrospective observational study with descriptive and analytical components that included all the patients admitted to the MSU at the CHP from 1 January 2017 to 31 December 2018. The sample comprised 116 patients, of which 55.2% (*n* = 64) had DDs, and 44.8% (*n* = 52) did not. Variables related to the patient sociodemographic profile, medical comorbidities, and their destination after discharge were collected from patient clinical histories for further study. Variables on addictions and other mental illnesses were obtained via clinical interviews by using the DSM-5 classification, while the *Clinical Global Impression* (CGI) [10] questionnaire was used to assess the overall patient clinical status. Data about the need for readmission or healthcare provision in the Emergency Department 3 and 6 months after discharge from the MSU was obtained from the patient medical histories. 

Statistical analysis was performed with SPSS software (version 25 for Windows, IBM Corp., Armonk, NY, USA). Pearson Chi-squared tests were used to compare the qualitative variables, and Student *t*-tests were employed to compare the data means. ANOVA was used to compare men and women with and without dual diagnosis (four groups). Once it had been verified that the necessary assumptions for its application were met, binary logistic regression was performed, applying the conditional forward method to check whether the variables studied could differentiate which individuals had DDs and which ones would require re-admission.

This research was authorized by the Ethics and Research Committee at the CHP and CEU Universities (CEI19/142). All the procedures implemented in contribution to this work complied with the 1975 Declaration of Helsinki, as revised in 2008.

## 3. Results

### 3.1. Sociodemographic and Clinical Characteristics

Of the 116 patients in our cohort, 40.5% were female, and 59.5% were male; their mean age was 40.6 years (*SD* = 12.6, and the range was 19–75 years). Table 1 shows the main patient characteristics according to the presence or absence of DDs. Of note, with respect to patients without DDs, patients with a DD were more frequently male (81.3%, *n* = 52 versus 32.7%, *n* = 17; *p* < 0.001); were younger (36.5 years versus 45.6 years; *p* < 0.001); and had experienced an earlier onset of mental illness (22.5 years versus 28.6 years; *p* = 0.002) (Table 1). Interestingly, the group of patients without DDs had more associated medical comorbidities (*p* = 0.004) (Table 1), especially cardiovascular and metabolic syndromes (data not shown).

Regarding the mental disorder diagnosis ($\chi^{2}$ = 22.89; *p* = 0.029), schizoaffective disorder was more frequent in patients with DDs (28.1%; *n* = 18 versus 9.6%; *n* = 5), and schizophrenia was more frequent in the group without DDs (55.8%; *n* = 29 vs. 37.5%; *n* = 24) (data not shown in tables). There were no differences in the rest of the diagnoses, with schizophrenia spectrum disorders predominating in both groups. The most frequent addiction in patients with DDs was the consumption of tetrahydrocannabinol (37.5%; *n* = 24), followed by polydrug addiction (31.3%; *n* = 20), alcohol consumption (15.6%; *n* = 10), and psychostimulants (15.6%, *n* = 10) (data not shown in tables).

### 3.2. Inpatient-Related Variables

Table 2 shows that there were no differences in the origin of the patients according to whether they presented a DD or not, with the most common origin being the Short-Term Hospitalization Unit (SHU) in both groups. In addition, there were no differences between the patients with and without DDs in terms of the number of previous admissions to the SHU (3.37 versus 3.73, respectively; *p* = 0.591) or the MSU (*p* = 0.693) (Table 2). However, the duration of the SHU admission before transfer to the MSU was shorter for patients with DDs than that of the group without DDs (23.54 versus 37.24 days, respectively; *p* = 0.025) (Table 2). Nonetheless, no differences were observed in the duration of their admission to the MSU.

Table 3 shows the differences in the overall number and mean doses of the main psychotropic drugs used to treat these patients. No differences were observed in the number of prescribed drugs in either group (3.23 versus 3.56 in patients with DDs; *p* = 0.183), although a higher dosage of long-acting injectable antipsychotics (quarterly doses) was noted for the DD group (447.06 mg (*n* = 17) versus 592.10 (*n* = 30); *p* = 0.008), and the use of aripiprazole was greater in the group without DDs (26.9% (*n* = 14) versus 9.4% (*n* = 6); *p* = 0.013) (Table 3).

### 3.3. Clinical Evaluation, Destination at Discharge, and Events up to 6 Months (Data Not Shown in Tables)

There were no differences between either group in terms of the CGI classification, at either admission ($\chi^{2}$ = 8.21; *p* = 0.145) or upon discharge ($\chi^{2}$ = 5.007; *p* = 0.415). At admission, both groups usually presented ‘moderate disease’ (DD = 54.7%; without DD = 47.1%), while, at discharge, most patients identified with the ‘much better’ category (DD = 46.9%; without DD = 36.7%). Patients with DDs had more frequently been referred to outpatient resources (54.7%; $\chi^{2}$ = 6.66; *p* = 0.010), while those without DDs had usually been referred to residential resources (69.2%). Specifically, patients with DDs had more often been referred to the CHP’s Severe Dual-Disorder Program (SDDP; 11.1%; *n* = 7; $\chi^{2}$ = 11.35; *p* = 0.003). 

Analysis of the events at 6 months in half of the sample (year 2018) showed no differences between either group in terms of their average number of visits to Emergency Department (*t* = 1; *p* = 0.326) or readmissions to the MSU (*t* = 0.97; *p* = 0.336). However, compared to the group without DDs, the mean number of readmissions to the SHU (*t* = 2.63; *p* = 0.012) were higher in patients with DDs (0.16 (*SD* = 0.37) versus 0 admissions, respectively). In addition to more frequent SHU re-admissions among patients with DDs (*n* = 6), further analysis revealed that these admissions were related to fewer previous admissions to the MSU (*t* = 7.57; *p* < 0.001), higher doses of olanzapine (*t* = 2.82; *p* = 0.009), a non-schizophrenia spectrum diagnosis ($\chi^{2}$ = 5.15; *p* = 0.023; especially affective spectrum diagnoses: schizoaffective disorder = 33.3%, bipolar disorder = 33.3%, and recurrent depressive disorder = 16.7%), a desvenlafaxine prescription ($\chi^{2}$ = 10.72; 0.001), medication with lithium ($\chi^{2}$ = 4.19; *p* = 0.041), and referral to outpatient resources ($\chi^{2}$ = 3.98; *p* = 0.046). Re-admission was not related to any of the other variables we studied. 

The only variable that showed association in the regression with the presence of DDs in the sample was the onset of mental illness (odds ratio = 0.55; 95% confidence interval = (0.31, 0.95); *p* = 0.034) (adjusted for age, sex, medical comorbidities, SHU admission duration before referral (days), and Quarterly LAID dose). None of the variables showed association in the regression with readmission to the SHU (data not shown in tables). 

### 3.4. Results Disaggregated by Sex

Table 4 shows the results according to the patient sex and presence or absence of DDs, comparing four groups: women with DDs, women without DDs, men with DDs, and men without DDs. Regarding drugs, aripiprazole had been prescribed more in women with DDs (34.3%; *n* = 12), and in a lower percentage in men without DDs (5.8%; *n* = 3; $\chi^{2}$ = 12.78; *p* = 0.005) (data not shown in tables). There were no differences between the four groups in terms of the percentage of prescriptions nor in the doses for any of the other drugs studied (data not shown). When the substances consumed were compared between women and men with DDs, women consumed a higher proportion of alcohol ($\chi^{2}$ = 10.77; *p* = 0.013; 41.7%, *n* = 5) while men consumed more tetrahydrocannabinol (44.2%; *n* = 23) (data not shown in tables).

## 4. Discussion

The profile of the patients in this study coincided with that of other MSUs [11], with a predominance of males without a partner and with lower education levels compared to the general population [12]. However, the mean age of 40.6 years at the time of admission to our MSU was relatively high compared to other reports. This was probably because of the late referral of patients to this hospital resource in our setting, suggesting that greater integration of these patients into existing resources is still required [13,14]. 

In addition, in line with patients from different settings [15], the prevalence of DDs in this present study was very high. In this work, the sociodemographic profile of the patients with DDs was similar to that of other studies [16,17], although our patients were generally younger and had a lower age at mental illness onset than those without addictions. As previously shown, this may be because drug use is a poor prognostic factor, and, furthermore, patients with earlier-onset mental illness required more prompt admission to the MSU [15,18,19].

In terms of the types of substances consumed, tetrahydrocannabinol was most often used, followed by polydrug addiction patterns. The latter had the greatest impact as a poor prognosis factor from among the variables we examined, which was also in line with various reviews [20,21]. In contrast, an associated medical comorbidities, especially cardiovascular and metabolic syndromes, was observed in more than half of the study cohort (52.6%). Of note, this finding concurs with several publications in which a greater excess of mortality from somatic causes was observed in psychiatric populations [22,23]. We believe that the higher prevalence of concomitant pathologies in patients without DDs may be related to the older mean age in this group in our study. 

Both patients with and without DDs had usually been referred from the SHU, with a few referrals from community outpatient resources. However, we believe that these referral rates, ideally, should be the opposite way around, as practiced in some other Spanish units [24]. This would facilitate early patient care provision which could help minimize the impact of these diseases. The mean admission duration to the SHU was 29.24 days, while patients with DDs required shorter admission times prior to their transfer to the MSU. It may be because (a) detoxification accelerates the improvement of psychopathology; (b) some MSU admissions were programmed prior to SHU admissions; or (c) admission acceptance to the MSU by the team is usually faster for more severe patient profiles. 

Polypharmacy psychopharmacological treatments stood out from among all the patient therapies applied. According to current trends to improve both adherence and help prevent relapses [25], long-acting injectable antipsychotics had been prescribed to 62.9% of our cohort. Likewise, the prescription of clozapine to 15.5% of our cohort was an indicator of good practice because this medication is considered one of the most effective atypical antipsychotics [26,27]. Although no differences were found between the two groups in terms of the percentage of clozapine prescribed, we understand that its prescription should be increased, as recommended by various publications [28,29]. Of note, patients with DDs generally required higher doses of quarterly depot paliperidone, which was in line with several other studies [5,30,31].

From a gender perspective, a higher proportion of men in our sample were affected by DDs compared to women. Furthermore, as also observed in other studies [32,33], there were no differences in the mean age and onset of mental illness between men and women with DDs. In addition, in agreement with other studies [34], we observed a higher percentage of women with comorbidities with a cluster B personality disorder.

Finally, we believe that the fact that few differences were found between the group with and without DDs was attributable to the higher level of global disease severity in our study sample, as indicated by the very high CGI categories of all the patients upon admission. To help improve the care provided, we propose that patients, especially those from the community, should be referred to the MSU earlier; in turn, in our opinion, patients from the SHU should only be referred after their first psychotic or serious affective episode. This would make it easier to keep patients with severe mental illnesses in the community and non-residential network, as achieved by the MSU in patients with DDs. 

Importantly, the efficacy of the approach used by the MSU was similar among patients with or without DDs, while it is worth highlighting the specific work carried out in areas, such as relapse prevention, in the latter. Strategies for close post-discharge follow-up of patients with DDs and on the affective spectrum must also be implemented because this group is most likely to be re-admitted to the SHU. Thus, promoting hospital structures and outpatient units similar to the SDDP in our hospital would facilitate close patient monitoring, in line with different national and international recommendations [35,36].

### Limitations

First, we were only able to analyze the follow-up in half of the sample in this work. Thus, because we only identified 6 patients in our study who had required re-admission to the SHU, our statistical analysis of this phenomenon was quite limited. Second, we followed the patients up at 6 months; hence, although we showed that the low global readmission rate was an indicator of the effectiveness of our MSU, the validity of the results in that specific analysis could be questioned. Third, given the exploratory nature of this study, potential confounders that biased the results may have been left out. Last, because of the retrospective nature of this study, we cannot infer the direction of the relationship between the variables, and prospective studies are needed for replication.

## 5. Conclusions

The MSU, considered as Transitional Residential Rehabilitation approach, could be considered as a psychosocial rehabilitation option for patients with severe mental disorders, regardless of the presence of DDs. The patients with DDs who were admitted to the MSU were more frequently male, younger, had required shorter admissions during previous hospitalizations, and required higher doses of depot paliperidone. Finally, the sex difference found in this work further supports the need to consider gender perspectives in the management of patients with DDs.

## Figures and Tables

**Table 1 brainsci-12-00024-t001:** Differences in the sociodemographic and clinical variables of patients with or without dual disorder (DDs).

	Without DD (*n* = 52)	DD (*n* = 64)		
	*n* (%)/ *M* (*SD*)	*n* (%)/ *M* (*SD*)	*t; p* (*β* − 1)	$\chi^{2}\left(df\right);\;p$
AGE	**45.60** (13.16)	**36.55** (10.66)	4.09; <0.001 (0.98)	
ONSET OF MENTAL ILLNESS	**28.65** (11.33)	**22.58** (8.68)	3.15; 0.002 (0.92)	
SEX				
Female	**35** (**67.3%**)	12 (18.8%)		28.06 (1); <0.001
Male	17 (32.7%)	**52** (**81.3%**)		
SPANISH NATIONALITY	47 (90.4%)	52 (81.3%)		1.91 (1); 0.167
MARITAL STATUS				
Single/Separated/Widowed	46 (88.5%)	58 (90.6%)		0.14 (1); 0.704
In a relationship	6 (11.5%)	6 (9.4%)		
INCOMPLETE PRIMARY EDUCATION	11 (21.2%)	23 (35.9%)		3.02 (1); 0.082
SOCIAL SITUATION				
Active	14 (26.9%)	23 (35.9%)		1.07 (1); 0.300
Disabled/Incapacity for work	38 (73.1%)	41 (64.1%)		
MEDICAL PATHOLOGY	**35** (**67.3%**)	**26** (**40.6%**)		8.19 (1); 0.004
SCHIZOPHRENIA SPECTRUM DIAGNOSIS	37 (71.2%)	48 (75%)		0.21 (1); 0.642

Note: DD: Dual Disorder. There were significant differences between the values marked in bold. Schizophrenia Spectrum Diagnosis included: Schizophrenia, Delusional Disorder, Unspecified Schizophrenia Spectrum and Other Psychotic disorder, and Schizoaffective Disorder.

**Table 2 brainsci-12-00024-t002:** Differences in the variables related to the admission of patients with or without dual disorder (DDs).

	Without DD (*n* = 52)	DD (*n* = 64)		
	*n* (%)/ *M* (*SD*)	*n* (%)/ *M* (*SD*)	*t*; *p* (*β* − 1)	$\chi^{2}(\mathrm{df});\;p$
REFERRAL TO THE				
SHU	32 (61.5%)	48 (75%)		2.42 (1); 0.119
OMHU/Others	20 (38.5%)	16 (25%)		
No. previous SHU admissions	3.37 (3.41)	3.73 (3.85)	0.53; 0.591 (0.13)	
No. previous MSU admissions	0.52 (0.70)	0.58 (0.86)	0.39; 0.693 (0.10)	
SHU admission duration before referral (days)	**37.24** (**35.42**)	**23.54** (**18.02**)	2.30; 0.025 (0.84)	
MSU admission duration (days)	192.87 (137.14)	154.71 (135.94)	1.45; 0.148 (0.41)	

Note: DD: Dual Disorder. There were significant differences between the values marked in bold. MSU: Medium-Stay Unit; SHU: Short-Term Hospitalization Unit; OMHU: Outpatient Mental Health Unit.

**Table 3 brainsci-12-00024-t003:** Differences in the dosage and number of psychotropic drugs used at Medium-Stay Unit discharge.

	Without DD (*n* = 52)	DD (*n* = 64)		
	Mean (*SD*)	Mean (*SD*)		*t*; *p* (*β* − 1)
Number of prescribed drugs	3.23 (1.46)	3.56 (1.20)		1.33; 0.183 (0.35)
	**%** (***n***) **Mean Dose** (***SD***)	**%** (***n***) **Mean Dose** (***SD***)	$\chi^{2}$ (***df***)**; *p***	***t*; *p*** (***β* − 1**)
ORAL ANTIPSYCHO-TICS				
Quetiapine	34.6 (18)	43.8 (28)	1 (1); 0.317	0.65; 0.515 (0.26)
	288.89 (259.83)	350 (335.54)		
Olanzapine	25 (13)	26.6 (17)	0.03 (1); 0.848	0.008; 0.994 (0.05)
	16.15 (8.20)	16.18 (7.18)		
Paliperidone	11.5 (6)	15.6 (10)	0.40 (1); 0.526	0.74; 0.468 (0.66)
	10 (5.25)	11.70 (3.86)		
Aripiprazole	**26.9** (14)	**9.4** (6)	6.19 (1); 0.013	0.91; 0.372 (0.75)
	14.64 (7.19)	19.17 (15.30)		
Clozapine	11.5 (6)	18.8 (12)	1.13 (1); 0.286	0.35; 0.729 (0.24)
	237.50 (105.77)	256.25 (106.66)		
Amisulpride	5.8 (3)	4.7 (3)	0.06 (1); 0.794	0.55; 0.607 (0.81)
	1066.67 (230.94)	1266.67 (577.35)		
LONG-LASTING INJECTABLE ANTIPSYCHOTICS				
Quarterly LAID	32.7 (17)	46.9 (30)	2.39 (1); 0.122	2.79; 0.008 (0.97)
	**447.06** (116.55)	**592.10** (238.46)		
Monthly LAID	11.5 (6)	12.5 (8)	0.02 (1); 0.874	1.18; 0.260 (0.95)
	200 (88.03)	150 (70.71)		
LAIA	9.6 (5)	10.9 (7)	0.05 (1); 0.816	0.77; 0.454 (0.78)
	420 (109.54)	385.71 (37.79)		
ANTIDEPRESSAN-TS				
Desvenlafaxine	7.7 (4)	12.5 (8)	0.71 (1); 0.398	1.02; 0.332 (0.95)
	187.50 (85.39)	143.75 (62.32)		
Mirtazapine	3.8 (2)	9.4 (6)	1.36 (1); 0.243	2.12; 0.078 (0.96)
	22.50 (10.60)	30 (0)		
Fluoxetine	5.8 (3)	1.6	1.52 (1); 0.217	0.86; 0.478 (0.99)
	30 (10)	20 (1)		
Escitalopram	1.9	3.1 (2)	0.16 (1); 0.685	1.44; 0.386 (1)
	5 (1)	30 (14.14)		
MOOD STABILISERS				
Valproate	13.5 (7)	1264.71/26.6	3.001 (1); 0.083	0.14; 0.889 (0.11)
	1228.57 (618.37)	(358.71; 17)		
Lithium	5.8 (3)	12.5 (8)	1.51 (1); 0.218	0.06; 0.950 (0.07)
	933.33 (230.94)	950 (410.57)		
BENZODIAZEPIN-ES				
Clonazepam	25 (13)	20.3 (13)	0.36 (1); 0.547	0.81; 0.423 (0.52)
	1.86 (1.00)	2.46 (2.41)		
Diazepam	11.5 (6)	9.4 (6)	0.14 (1); 0.704	0.91; 0.381 (0.88)
	10.83 (4.91)	15.83 (12.41)		
Clorazepate	9.6 (5)	9.4 (6)	0.002 (1); 0.965	1.55; 0.178 (0.99)
	98 (60.88)	52.50 (26.02)		

Note: DD: Dual Disorder. There were significant differences between the values marked in bold. LAID: long-acting injectable paliperidone, LAIA: long-acting injectable aripiprazole.

**Table 4 brainsci-12-00024-t004:** Results disaggregated by sex and the presence or absence of dual disorders.

	Female, No DD (*n* = 35) *M* (*SD*)/ % (*n*)	Female, DD (*n* = 12) *M* (*SD*)/ % (*n*)	Male, No DD (*n* = 17) *M* (*SD*)/ % (*n*)	Male, DD (*n* = 52) *M* (*SD*)/ % (*n*)	$F/\chi^{2}$ (*p*) Turkey/RTC
Age	**46.43** (**11.76**)	38.50 (10.79)	43.88 (15.91)	**36.10** (**10.68**)	5.83 (0.001) 0.001
Onset of mental illness	**30.65** (**11.23**)	**20.33** (**7.53**)	24.65 (10.74)	**23.10** (**8.91**)	5.28 (0.002) 0.012/0.004
Spanish nationality	88.6 (31)	91.7 (11)	94.1 (16)	78.8 (41)	3.47 (0.324)
MARITAL STATUS:					
Single/Separated/Widowed	85.7 (30)	75 (9)	94.1 (16)	94.2 (49)	4.90 (0.179)
In a relationship	14.3 (5)	25 (3)	5.9 (1)	5.8 (3)	
Incomplete primary education	17.1 (6)	25 (3)	29.4 (5)	38.5 (20)	4.71 (0.194)
SOCIAL SITUATION:					
Active	31.4 (11)	33.3 (4)	17.6 (3)	36.5 (19)	2.12 (0.548)
Disabled/Incapacity for work	68.6 (24)	66.7 (8)	82.4 (14)	63.5 (33)	
Medical pathology (yes)	**68.6** (**24**)	50 (6)	64.7 (11)	**38.5** (**20**)	8.78 (0.032) 2.3/−0.2/1.1/−2.7
REFERRAL TO THE MSU					
SHU	68.6 (24)	66.7 (8)	47.1 (8)	76.9 (40)	5.38 (0.146)
OMHU/Others	31.4 (11)	33.3 (4)	52.9 (9)	23.1 (12)	
Diagnosis of axis I schizophrenia spectrum disorder	65.7 (23)	50 (6)	82.4 (14)	80.8 (42)	6.54 (0.088)
Diagnosis axis II cluster B disorder	25.7 (9)	**58.3** (**7**)	11.8 (2)	25 (13)	8.16 (0.043) −0.2/2.6/−1.5/−0.4
Previous SHU admission	3.23 (3.28)	2.67 (1.77)	3.65 (3.77)	3.98 (4.16)	0.56 (0.642)
SHU admission duration before referral	**43.14** (**35.58**)	22.82 (18.03)	25.43 (33.18)	**23.71** (**18.20**)	3.61 (0.016) 0.013
Previous MSU admission	0.46 (0.70)	0.50 (0.79)	0.65 (0.70)	0.60 (0.89)	0.31 (0.817)
Days of admission to MSU	206.45 (138.12)	199.08 (219.89)	166.56 (135.65)	144.48 (108.76)	1.53 (0.210)
Number of drugs	3.34 (1.53)	4.08 (1.08)	3 (1.32)	3.44 (1.21)	1.62 (0.188)
CGI AT THE TIME OF ADMISSION:					
‘Slightly ill’	0 (0)	0 (0)	0 (0)	5.8 (3)	3.73 (0.292)
‘Very ill’	100 (35)	100 (12)	100 (16)	94.2 (49)	
CGI AT DISCHARGE:					
‘Much better’	96.8 (30)	100 (12)	93.3 (14)	100 (51)	3.45 (0.327)
‘No change’	3.2 (1)	0 (0)	6.7 (1)	0 (0)	
DESTINATION UPON DISCHARGE:					
Residential resource	65.7 (23)	66.7 (8)	76.5 (13)	**40.4** (**21**)	9.93 (0.019) 1.4/0.8/1.8/−3.1 −1.4/−0.8/−1.8/3.1
Outpatient resource	34.3 (12)	33.3 (4)	23.5 (4)	**59.6** (**31**)	
DESTINATION UPON DISCHARGE:					
OMHU	46 (14)	41.7 (5)	26.7 (4)	**60.8** (**31**)	14.82 (0.022) −0.4/−0.6/−1.9/2.1 −1.7/1.5/−1.1/1.3 1.3/−0.1/2.5/−2.8
SDDP	0 (0)	16.7 (2)	0 (0)	9.8 (5)	
Residential	53.3 (16)	41.7 (5)	**73.3** (**10.2**)	**29.4** (**15**)	
Emergency Department attendance after discharge	1.37 (3.83)	1.83 (2.85)	1.33 (3.26)	0.35 (0.87)	1 (0.400)
Readmissions to the SHU after discharge	0 (0)	0.33 (0.51)	0 (0)	0.13 (0.33)	2.42 (0.075)
Readmissions to the MSU after discharge	0.05 (0.22)	0 (0)	0.17 (0.40)	0.03 (0.17)	0.77 (0.515)

Note: There were significant differences between the values marked in bold. DD: dual disorder; MSU: Medium-Stay Unit; SHU: Short-Term Hospitalization Unit; OMHU: Outpatient Mental Health Unit; SDDP: Severe Dual-Disorder Program; CGI: Global Clinical Impression classification. Schizophrenia Spectrum Diagnosis included: Schizophrenia, Delusional Disorder, Unspecified Schizophrenia Spectrum and Other Psychotic disorder, and Schizoaffective Disorder.

## Data Availability

The datasets generated during and/or analyzed during the current study are available from the corresponding author on reasonable request.

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
