# Peer review of "Addressing Dual Disorders in a Medium-Term Admission Unit"

_brainsci, 2021, doi:10.3390/brainsci12010024_

Round 1
Reviewer 1 Report
The subject of this paper is of interest. In psychiatry, RCTs and other research often focus on one disorder excluding comorbidity, while in fact multiple diagnoses are relatively prevalent. In addition, it is difficult to find mental health care for patients with severe mental illness as well as substance use disorder (they fall between two stools).
However, the present paper is nothing more than a description of the presence of DP in the MSU in a retrospective small study.
- The retrospective study design has disadvantages.
This design is at best hypothesis generating. More prospective studies are needed for replication.
- Statistical analyses and results: degrees of freedom
Chi-square tests do not only have the statistic chi-square and a p-value, but also a “degrees of freedom”. Every result of a chi-square test should be accompanied by the degrees of freedom. In addition, the t-test also has degrees of freedom, but it is less important here.
In addition, to me it is unclear which results in the text are also presented in the table and which not. I would advise to add either add e.g. “(table 1)” or “(data not shown)” to all results.
- Heading “income-related variables”
The heading “income-related variables” does not reflect the content of the paragraph. Table 2 and table 3 are discussed; i.e. MSU/SHU, previous admissions, duration of admission (table 2) and dosage of drugs (table 3).
- Logistic regression results
Age of symptom onset was associated with DP. I would advise the authors not to use terminology such as “predictive power”. The authors simply analysed associations, not prediction models. In addition, although dichotomisation leads to loss of information, I wonder whether the association between age of onset and probability of DP is linear. If not linear, forcing the model to analyse as if linear gives invalid results.
- Confounding and interaction
The authors only perform crude analyses, using t-tests and chi-square tests. Were there no hypotheses for confounding? In addition, the authors performed logistic regression. This type of analysis enables inclusion of confounders. However, the statistical analysis does not provide any information on this. Were there again no hypotheses for confounders?
In the last heading of the results (“Disaggregated” by sex), the authors stratify by sex. This was not announced in introduction or statistical analysis. The hypothesis for differences between males and females should be provided before analysing. If there is a hypothesis that warrants analysis, interaction between sex and the other independent variables should be modelled. Perhaps numbers are too low to analyse this, but without this analyses conclusions whether there are differences between males and females cannot be drawn and thus stratification by sex is useless.
- Medical pathology
A large variety of variables is obtained from the medical records and described in the two groups. Which variables were included, what they actually mean and what the hypothesis is to include them in the analysis is unknown. I would like to know
(1) What the variable “medical pathology” actually means.
(2) Why schizophrenia spectrum diagnosis is in table 1 and other diagnoses are not.
(3) What the variables in table 3 mean, knowing that patients are included because they were admitted to the MSU. Is the first row referral after discharge?
- Table 2 misses a line
Table 2 has rows for MSU, SHU and OMHU/others. However, only 2 lines of results were presented.
- Paliperidone
Multiple antipsychotics (AP) are analysed. Only depot paliperidone is significantly different between the groups. The authors state that this is in line with other research, but this should be specified. Is paliperidone the only AP where this is plausible and reported earlier and why is that not mentioned in the introduction. Or could it just as easily have been any of the other AP while these were not significant. In that case, I would think it is a chance finding.
- Conclusion
That MSU is an effective treatment cannot be concluded from the present descriptive study.
Minor and language
- Language
The language is not fully adequate and should be checked by a translator.
- Sentence in discussion
Line 153: “In addition, in line with studies in other resource types [11], the …”
“other resource types” is not crystal clear. Looking at reference 11, I assume the authors mean “patients from different settings”.
- Tables 1-3 and power
- Tables 1 to 3 are difficult to read. I realise the authors want to present all information without taking too much space. Presenting results from 2 different statistical tests on one row of the table, 1 for the continuous measure and one for the percentages, is challenging. The figures per cell already are often presented in two lines within the row. I would advise to consequently present the percentage and the accompanying chi-square (with df) on the first line within the row and the mean, standard deviation with accompanying t-test on the second line. The first column presenting totals can be removed.
- Ability to read the tables is important because the number of patients per group are relatively low and, thus, power is low. Differences that are not statistically significant may be clinically relevant. The chi-squares in the text that are not presented in the tables are, therefore, difficult to interpret (CGI, diagnosis).
- The two columns after the total are presenting results for the group with DP and the group without. In the tables those 2 columns do not have meaningful headings. So, only from the numbers I can decide which group is DP and which not.
- What is the unit of admission duration in table 2?
- Line 173/174 “The mean admission duration to the SHU was 29.24 days, perhaps because the cases of patients who require hospital rehabilitation were more serious and therefore required longer admissions.” I cannot follow this sentence. The sentence starts with an admission duration in the total group. If I understand correctly the rest of the sentence is interpretation, but because only the 29 days in the total group are mentioned, I do now know what is explained and why.
- Onset of mental illness / disease start age
The authors use the term “onset of mental illness” in the text but in table 1 they use “disease start age”. I advise to use the same terminology throughout the paper.
Author Response
Dear Reviewer 1,
First of all, thank you very much for your appreciations and constructive criticism of our article.
Please find enclosed in a word document the answers to your suggestions.
Kind regards

Reviewer 2 Report
I think this is a very well written article that covers very important subject, as all data sets we can receive about dual diagnoses are important. The article is not novel, however, it covers perfectly basic information from Spain. I think it will be a very good fit for Brain Sciences, however, I would like to suggest one tiny change: please elaborate about Medium Stay Units etc. - how do those work in Spain? What are the other types of psychiatric units? What is the organization of hospital-based mental health care there? Paragraph in the introduction would be enough. Please remember that Brain Sciences is an international journal and that's why such info would be a value in the introduction section.
Author Response
Dear Reviewer 2,
First of all, thank you very much for your appreciations and constructive criticism of our article.
Please find enclosed in a word document the answers to your suggestions.
Kind regards
Round 2
Reviewer 1 Report
The author incorporated most of my comments, but some problems remain to be addressed.
- This design is at best hypothesis generating. More prospective studies are needed for replication.
I am happy that the authors agree. However, this should also be acknowledged in the paper. E.g. in the limitations section in the discussion.
- logistic regression.
Good to know that the association is linear. Not only me as the reviewer, but also the reader should know that.
- Confounding
I am happy that the authors included confounders. However, they should not choose confounders based on significances in the crude analyses. Those crude analyses can also be confounded. Instead, they should reason what could be confounders in theory add them and leave them in the analysis (no stepwise). Those confounders should be provided in the statistical analysis paragraph.
If the authors chose a different strategy because this is an explorative study, they should discuss the disadvantages in the limitations section.
- Interaction
That the authors now adapted the aim is an improvement. However, “gender perspective” is not crystal clear and the reason for analysing interaction is still not described in the introduction. If the word count for this brief report is not enough to include this, the authors can better remove interaction from their paper.
- Variables: Schizophrenia spectrum diagnosis.
I agree that not providing all columns safes space, but then the authors should at least provide a footnote with Schizophrenia spectrum diagnosis.
- Conclusion.
“The MSU has proven to be a rehabilitative treatment option for patients with severe mental disorders”.
My comment remains the same: “this cannot be concluded from the present study”. This sentence should be either removed or it needs a reference.
Author Response
Dear Reviewer,
We are glad we were able to incorporate most of the recommendations and welcome new suggestions.
Then we go on to answer them, we consider that they have also already been incorporated in this latest version.
- This design is at best hypothesis generating. More prospective studies are needed for replication.
We have added this limitation in line 254.
- Logistic regression.
We have clarified it in lines 76-77.
- Confounding
We have added this limitation in line 251-252.
- Interaction
We consider the gender perspective essential, so we have decided not to do without their interactions in the article.
- Variables: Schizophrenia spectrum diagnosis.
We have added all the diagnoses in a footnote at tables 1 and 4.
We have reformulated the conclusion, taking into account the opinion of the reviewer (lines 256-267).
Yours sincerely,
